# Correlation between Subchondral Insufficiency Fracture of the Knee and Osteoarthritis Progression in Patients with Medial Meniscus Posterior Root Tear

**DOI:** 10.3390/diagnostics13233532

**Published:** 2023-11-26

**Authors:** Bing-Kuan Chen, Yi-Cheng Lin, Yu-Hsin Liu, Pei-Wei Weng, Kuan-Hao Chen, Chang-Jung Chiang, Chin-Chean Wong

**Affiliations:** 1Division of General Medicine, Department of Medical Education, Taipei Medical University Shuang Ho Hospital, New Taipei City 23561, Taiwan; 22271@s.tmu.edu.tw; 2Department of Orthopedics, Taipei Medical University Shuang Ho Hospital, New Taipei City 23561, Taiwan; yichenglin@tmu.edu.tw (Y.-C.L.); wengpw@tmu.edu.tw (P.-W.W.); cjchiang@tmu.edu.tw (C.-J.C.); 3Department of Orthopedics, School of Medicine, College of Medicine, Taipei Medical University, Taipei 11031, Taiwan; 4International Ph.D. Program in Biomedical Engineering, College of Biomedical Engineering, Taipei Medical University, Taipei 11031, Taiwan; 5Research Center of Biomedical Devices, Taipei Medical University, Taipei 11031, Taiwan; 6School of Biomedical Engineering, College of Biomedical Engineering, Taipei Medical University, Taipei 11031, Taiwan; 7Graduate Institute of Biomedical Materials and Engineering, Taipei Medical University, Taipei 11031, Taiwan; 8International Ph.D. Program for Cell Therapy and Regenerative Medicine, College of Medicine, Taipei Medical University, Taipei 11031, Taiwan

**Keywords:** medial meniscus, root tear, subchondral insufficiency fracture of the knee, arthroscopic, pullout repair

## Abstract

A medial meniscus posterior root tear (MMPRT) contributes to knee joint degeneration. Arthroscopic transtibial pullout repair (ATPR) may restore biomechanical integrity for load transmission. However, degeneration persists after ATPR in certain patients, particularly those with preoperative subchondral insufficiency fracture of the knee (SIFK). We explored the relationship between preoperative SIFK and osteoarthritis (OA) progression in retrospectively enrolled patients who were diagnosed as having an MMPRT and had received ATPR within a single institute. Based on their preoperative magnetic resonance imaging (MRI), these patients were then categorized into SIFK and non-SIFK groups. OA progression was evaluated by determining Kellgren–Lawrence (KL) grade changes and preoperative and postoperative median joint widths. SIFK characteristics were quantified using Image J (Version 1.52a). Both groups exhibited significant post-ATPR changes in medial knee joint widths. The SIFK group demonstrated significant KL grade changes (*p* < 0.0001). A larger SIFK size in the tibia and a greater lesion-to-tibia length ratio in the coronal view were positively correlated with more significant KL grade changes (*p* = 0.008 and 0.002, respectively). Thus, preoperative SIFK in patients with an MMPRT was associated with knee OA progression. Moreover, a positive correlation was observed between SIFK lesion characteristics and knee OA progression.

## 1. Introduction

The meniscus is a curved, fibrocartilaginous structure in the knee joint with various functions, including lubrication, proprioception, nutrient supply, and load distribution across the joint [1]. Moreover, the meniscus roots play a crucial role in converting the tibiofemoral axial load into circumferential hoop stress [2,3,4], and meniscus root injury can lead to further damage.

A meniscal root tear, defined as an avulsion injury of the root insertion or complete radial tears within 1 cm of the meniscus root attachment [5,6], is a specific meniscus root injury type. Most patients with this injury present with a sharp snap in their knee after descending knee movements and high knee flexion activities without preceding trauma [7,8,9]. Of all meniscus roots, the medial meniscus posterior root is the most vulnerable to injury because it is the least mobile but transmits the greatest pressure [10].

Medial meniscus posterior root tears (MMPRTs), accounting for 10–21% of all arthroscopic meniscal surgeries, affect 100,000 patients annually [6,11]. With the disruption of the biomechanical integrity, the MMPRT contributes to degenerative articular wear and accelerated arthritic changes in the knee joints [10,12,13].

Over the past 20 years, various treatment options, including nonsurgical treatment, meniscus root repair, and partial meniscectomy, have been proposed for MMPRTs. Of these treatment options, meniscus root repair provides better clinical outcomes, including higher tissue healing and clinical performance as well as a lower rate of conversion to joint replacement, through the surgical restoration of the knee joint biomechanical integrity [14,15,16,17,18]. Furthermore, partial meniscectomy is associated with an increased incidence of chondrolysis for lateral meniscus tears and osteoarthritis (OA) for medial meniscus tears [19]. Hence, meniscus root repair is considered the mainstream modality for meniscus root tears.

Suture anchor fixation and arthroscopic transtibial pullout repair (ATPR) are the two most adopted surgical techniques for meniscus root repair. Both techniques have been reported to effectively restore medial root attachment and prevent structural meniscal extrusions [9]. Compared with suture anchor fixation, ATPR is associated with a higher survival rate and short-to-medium-term clinical performance [20,21,22]. From a surgical perspective, ATPR is also associated with lower risks of neurovascular injuries and implant-related complications, such as anchor loosening [10,23,24,25,26]. Therefore, compared with suture anchor fixation, ATPR is more commonly employed for meniscus root repair [27].

A subchondral insufficiency fracture of the knee (SIFK), which was formerly referred to as a stress fracture, can occur as a complication of a posterior root tear of the medial meniscus. This type of fracture is typically observed in the weight-bearing region of the medial femoral condyle [7,28,29]. SIFK contributes to fluid accumulation in the bone marrow, leading to elevated intraosseous pressure and local osteonecrosis [30,31].

Studies have indicated that approximately 30% of patients with SIFK undergo conversion into arthroplasty [32,33]; thus, SIFK may be a poor prognostic factor of knee OA progression. Medial meniscus root repair can cease SIFK progression and knee-degenerative changes [5,34]. However, the effects of SIFK on patients with an MMPRT undergoing ATPR remain unknown.

In the current single-center retrospective study, we hypothesized that a preoperative magnetic resonance imaging (MRI)-detected SIFK and its characteristics are prognostic factors of knee OA progression in MMPRT patients who have undergone ATPR.

## 2. Materials and Methods

### 2.1. Study Design and Participants

This single-center, retrospective comparative study investigated the correlation between preoperative SIFK and knee OA progression in patients with persistent knee pain who were diagnosed as having an MMPRT and had undergone ATPR. All procedures, hospital stay, and clinic visits occurred at the Orthopedics Department of Shuang-Ho Hospital (New Taipei City, Taiwan). All patients who underwent ATPR from March 2019 to January 2023 were eligible for inclusion in this study.

The inclusion criteria were (1) MMPRT with extrusion, as revealed by MRI; (2) root tears with 1–2 Kellgren–Lawrence (KL)-grade osteoarthritic changes on radiographs; (3) type 2 MMPRT according to Laprade classification; and (4) maximum symptom duration of 6 months. Our exclusion criteria were (1) secondary osteonecrosis induced by iatrogenic events (e.g., long-term steroid use or radiation exposure), (2) previous surgical interventions noted on MRI or radiographs, (3) root tears with 3-4 KL-grade osteoarthritic changes, and (4) missing data or loss to follow-up.

The included patients were retrospectively categorized into SIFK and non-SIFK groups according to preoperative MRI scans of their affected knees (Figure 1). In both groups, outcome measurements were performed at the postsurgical follow-up visit.

### 2.2. Patient Demographics and Preoperative and Postoperative Outcome Assessments

#### 2.2.1. Patient Demographics

Patient information, including sex, height, weight, body mass index (BMI), and accurate age on the operative day, was collected. Follow-up duration was defined as the duration between the dates of the latest preoperative and postoperative knee anteroposterior (AP) standing radiography follow-ups.

#### 2.2.2. Image-Based MMPRT Diagnosis

In this study, MMPRT was diagnosed mainly on the basis of positive MRI findings. A slope-like swelling of the medial meniscus posterior segment, resembling a giraffe neck in the coronal view (i.e., giraffe neck sign), indicates the presence of an MMPRT [35,36,37]. Patients with an MMPRT also demonstrate a truncation sign, which is defined as a vertical linear root defect in the midcoronal view (i.e., cleft sign); it is correlated with medial meniscus extrusion of >3 mm [13,38]. Furthermore, in patients with an MMPRT, the absence of an identifiable meniscus in the sagittal view (i.e., ghost sign) indicates the presence of a radial tear [10,35]. Patients with an MMPRT may also exhibit a fluid gap and complete discontinuity of the posterior root (i.e., radial tear sign) [35]. In addition to the medial extrusion sign, more than 90% of patients with an MMPRT exhibit any two of the aforementioned MRI signs [36].

#### 2.2.3. SIFK Identification

SIFK is the most effectively visualized on T2-weighted or proton-density-weighted MRI scans. These MRI scans display the following characteristics: (1) a curvilinear or a parallel hypointense line or region adjacent to the subchondral bone plate, (2) thickened subchondral plate indicating callus and granulation formation, (3) flattening or depression of the subchondral bone plate, and (4) a fluid-filled cleft underlying the subchondral bone plate [39]. Three experts independently selected the lesion and calculated the lesion size and lesion-to-tibia length using ImageJ (Version 1.52a; National Institutes of Health) [40]. The average lesion sizes and lesion-to-tibia length were also calculated and analyzed thereafter.

Here, the lesion sizes of SIFKs in the tibia and femur were measured on the basis of the maximum values measured in the coronal and sagittal MRI views, as follows:The length units were calibrated using the 1 cm scale bar in the MRI images;An outline was drawn along the subchondral plate horizontally, and the flattened and depressed parts were connected;An outline was drawn along the thickened subchondral plate perpendicularly, without ill-defined areas, such that a radiating, disrupted linear pattern was obtained (Figure 2).

Next, we calculated the lesion-to-tibia length ratio by dividing the horizontal lesion length by the horizontal distance from the medial condyle to the adjacent peak of the tibial spine, as well as the greatest coronal lesion size, shown on MRI images (Figure 3).

#### 2.2.4. Knee OA Progression Grading

The knee OA progression degree was quantified on the basis of the difference in KL grades evaluated on knee AP standing radiographs of the affected joint preoperatively and postoperatively [41]. The medial knee joint width, defined as the distance from the apex of the medial femur condyle to the posterior end of the tibia, was also measured [42].

After they were anonymized, all radiographs were independently reviewed by three orthopedic surgeons who were not a part of the current study. In case of disagreements, a consensus was reached after discussion.

### 2.3. ATPR for MMPRTs

An orthopedic surgeon (C.-C.W.) experienced in arthroscopic surgery and sports medicine performed all surgical procedures for ATPR by employing standard operative procedures. In brief, after general or spinal anesthesia, patients underwent surgery in the supine position. Standard anterolateral and anteromedial portals were established, and normal saline was then introduced into the joint. Visualization was achieved using a 30° arthroscope (Smith & Nephew, London, UK). A thorough knee examination was performed to evaluate the meniscal root tear site and any associated chondral and ligamentous injuries. The tear pattern or the affected medial meniscus root was carefully examined. Next, the cartilage of the planned root reattachment site on the tibial plateau was curetted to create a bone bed for improved tissue healing. Using a grasper, the meniscus was pulled to ensure that it would be reduced to its newly created footprint.

Next, a 2 cm incision was made longitudinally just medial to the tibial tubercle, and a transtibial tunnel was drilled using an anterior cruciate ligament-aiming device with a cannulated sleeve (Arthrex, Naples, FL, USA), targeted at the planned root reattachment site. A 4.5 mm cannulated reamer was then used to widen the tunnel. A self-capture suture passing device (Knee Scorpion; Arthrex, Naples, FL, USA) was used to place sutures on the far posterior portion of the medial meniscus with a No. 0 nonabsorbable braided suture (Fiberwire; Arthrex, Naples, FL, USA). In all cases, two loop stitches were placed approximately 5 mm medial to the lateral edge of the tear site. Subsequently, the meniscal sutures were pulled through the tibial tunnel by using a nitinol passing wire. Finally, by tensioning the free ends of the sutures, the meniscus root was reduced to the created footprint, and tibial fixation was performed using a 4.75 mm SwiveLock anchor (Arthrex, Naples, FL, USA); (Figure 4).

### 2.4. Postoperative Rehabilitation Protocol

Postoperatively, all patients were strictly required to maintain a non-weight-bearing condition in their injured limb for 4 weeks. Moreover, all patients received a home rehabilitation program including active knee range of motion (ROM) exercises, isometric muscle training, and quadriceps strengthening.

### 2.5. Statistical Analysis

All statistical analyses were performed using SPSS (version 25.0; IBM, Armonk, NY, USA) and GraphPad Prism (version 9.5.1.733; GraphPad). Descriptive data are presented as means ± standard deviations (SDs) or as numbers and frequencies. A *p* value of <0.05 was considered to indicate statistical significance.

We used the independent-sample *t* test to assess between-group differences in patient demographics, preoperative and postoperative medial knee joint widths, and follow-up durations. Moreover, we employed Fisher exact and Pearson chi-square tests to analyze between-group differences in sex distribution, preoperative and postoperative KL grades, and knee OA progression. Next, we used paired *t* and Wilcoxon signed-rank tests to assess within-group differences in the preoperative and postoperative medial knee joint widths and KL grades, respectively. Furthermore, independent-sample *t* and Mann–Whitney *U* tests were to determine between-group differences in postoperative medial knee joint space improvement and KL grade changes in the affected knee, respectively.

Next, the Kruskal–Wallis test was used to compare the distribution of lesion characteristics on MRIs among different subgroups. If statistical significance was noted in the Kruskal–Wallis test, multiple pairwise comparisons were performed using the Bonferroni correction to adjust for significant values. Finally, Fisher exact *t* and Pearson chi-square tests were used to compare the distribution of lesion sites in different subgroups.

## 3. Results

Of all 55 eligible patients, 7 were excluded because of loss to follow-up. Of the remaining patients, 20 and 28 patients were included in the non-SIFK and SIFK groups, respectively. As listed in Table 1, neither the patient characteristics nor the preoperative demographic data differed significantly between the two groups.

In both groups, an MMPRT mostly occurred in individuals who were middle-aged (mean age = 58.64 ± 9.94 years), female (67%), or overweight (mean BMI = 27.95 ± 5.25). Patients in both groups had similar preoperative KL grades. The mean follow-up durations were 11.25 ± 9.13 and 10.74 ± 6.88 months in the non-SIFK and SIFK groups, respectively.

### 3.1. Knee OA Progression

As illustrated in Figure 5A, significant medial knee joint width changes were noted in the affected knee after the indexed surgical procedure in both the SIFK (*p* < 0.001) and non-SIFK (*p* = 0.047) groups. Furthermore, intergroup comparisons indicated that postoperative medial knee joint width narrowing was significantly smaller in the non-SIFK group than in the SIFK group (*p* = 0.049).

As displayed in Figure 5B, significant changes in the median KL grade of the affected knee postoperatively indicated OA progression in the SIFK group (*p* < 0.0001), but not in the non-SIFK group (*p* = 0.0703). Furthermore, the SIFK group demonstrated higher knee OA progression postoperatively than the non-SIFK group (*p* = 0.005).

To quantify knee OA progression severity, we assessed the distributions of different KL grade changes between the SIFK and non-SIFK groups (Figure 6). Significantly more patients developed larger KL grade changes postoperatively in the SIFK group than in the non-SIFK group (*p* = 0.020).

### 3.2. Influence of Lesion Characteristics

Table 2 lists all lesion characteristics, including the lesion location, MRI view, and KL grade differences. Among the subgroups, significant differences were found in the median tibia lesion size in both the coronal (*p* = 0.008) and sagittal (*p* = 0.031) views. Furthermore, we observed a significant difference in lesion-to-tibia length ratios (*p* = 0.002). However, after Bonferroni correction, no significant between-group differences were noted in the tibial lesion size in the sagittal view. Most lesions were found in the medial compartment of the knee, and lesions extending to both the medial femur and medial tibia were found in 17 of 28 patients with SIFK. However, no significant differences were noted in the distribution of lesion sites among different subgroups.

## 4. Discussion

In the current study, the major finding is that in patients with an MMPRT receiving ATPR, preoperative SIFK was associated with knee OA progression. In particular, after the initial MMPRT diagnosis, patients diagnosed as having a preoperative SIFK had increased OA progression in terms of joint space narrowing and KL grade changes in the affected knee.

Studies have indicated the importance of accurately detecting an MMPRT and providing subsequent early surgical repair for the preservation and overall health of the affected knee joint. In a retrospective study including 52 patients with an untreated MMPRT, nonoperative treatment led to poor outcomes at the 10-year follow-up or later; the outcomes included an increase in the total knee arthroplasty conversion rate, significant KL grade progression, and a high mean visual analog scale score (=4.4) [43].

Kim et al. reported that the postoperative MRI of surgically treated patients displayed a decrease in not only the meniscus extrusion, but also the gap distance of their meniscus root tears. Notably, most patients had improved functional outcomes over ≥2 years of follow-up, without any significant changes in the KL grade of the affected knee. These results suggest the therapeutic efficacy of surgical root tear repair in restoring normal kinematic joint functions [20]. By contrast, a meta-analysis compared the clinical outcomes of MMPRT patients who underwent either partial meniscectomy or ATPR, and noted that ATPR was more effective than meniscectomy in improving patients’ clinical outcomes and survival. In particular, the ATPR group had a lower incidence of total knee arthroplasty conversion and radiographic OA progression over >5 years of follow-up [44]. Taken together, these findings—consistent with the main findings of the current study—demonstrate the therapeutic efficacy of ATPR in MMPRT management.

To the best of our knowledge, research identifying the preoperative predictors of knee OA progression after root repair is limited. To fill this gap, here, we have revealed the correlations of knee OA progression severity with the SIFK lesion size, including tibial lesion size in the coronal view and the lesion size-to-tibial spine length ratio.

SIFK is a subchondral plate fracture surrounded by perifocal “flame-like” bone marrow edema, which can extend along and beyond the adjacent epiphysis [45]. In the present study, the SIFK imaging characteristics of a patient population comprising 32 women and 16 men were investigated. Medial meniscus extrusion, a common MMPRT manifestation, is positively correlated with femur bone marrow edema volume and SIFK in women; this suggests the presence of a relationship between SIFK and an MMPRT [46]. Studies have also demonstrated that preexisting OA may increase the load of the compartment, leading to degenerative injuries of the meniscus and cartilage and finally contributing to SIFK development [29]. Studies have reported that fluid accumulation in the bone marrow can lead to focal ischemia, subsequent necrosis, and further knee OA progression [28,31]. Burr et al. reported that microcracks in the subchondral bone induce secondary ossification center reactivation, resulting in cartilage loss and subsequent degenerative changes [32].

In a retrospective study including 249 MMPRT patients with SIFK, SIFK in the medial femoral condyle was considered a predictive factor for arthroplasty [47]. Moreover, Rozing et al. reported that OA tends to develop in knees with SIFK of lesion size > 2.3 cm^2^ [48]. By contrast, Walsh et al. demonstrated a positive correlation of the medial bone marrow lesion size with cartilage loss, which ultimately leads to the degeneration of the affected knee [49]; this result corroborates the present findings. The aforementioned studies have indicated that the SIFK lesion size may play a crucial role in OA progression, particularly in patients with an MMPRT. In the present study, although patients without SIFK demonstrated a slight narrowing of the medial joint width postoperatively compared to preoperatively (*p* = 0.047), no significant KL grade change was identified after ATPR. By contrast, patients with SIFK demonstrated significant joint space narrowing and OA progression (Figure 5 and Figure 6). We believe that this discrepancy in disease progression between the SIFK and non-SIFK groups is attributable to the role of SIFK in knee OA pathogenesis in patients with an MMPRT. In the future, new treatment strategies or combined procedures conducive to knee joint preservation should be considered for treating patients with an MMPRT and concomitant SIFK.

Because it increases medial OA progression and leads to unfavorable Lysholm score risks, >5° mechanical varus malalignment is considered a crucial predictor of poor surgical outcomes in patients with an MMPRT [50]. Given the roles of malalignment and the increased tibiofemoral pressure in meniscus injury and knee OA development, high tibial osteotomy (HTO) can effectively shift the mechanical loading from the affected compartment to the more preserved region of the knee joint [51,52,53]. An ideal HTO candidate is a younger (<65-year-old) individual with isolated medial OA, notable varus deformity of the knee, adequate knee joint ROM without contracture, and no knee ligamentous instability [54,55]. HTO may be an optimal choice for SIFK treatment. Choi et al. reported significant improvements in the MRI Osteoarthritis Knee Score grade (MOAKS; a scoring system that considers bone marrow lesion location and size) in both patients with primary OA and those with OA related to SIFK after open-wedge HTO [56]. Furthermore, a study including synovial fluid analysis demonstrated a significant decrease in the levels of proinflammatory cytokines, including interleukin (IL) 6, IL-8, and matrix metalloproteinase 13, in patients who received HTO. Taken together, the aforementioned studies have revealed not only the biomechanical role but also the biological effects of the load-shifting procedure in ameliorating the clinical progression of cartilage and osteochondral injuries, thus providing crucial information for the future clinical judgments of MMPRT patients with concomitant SIFK [57]. Additional studies, particularly high-quality randomized controlled trials, for accurately delineating the efficacy of combined HTO and MMPRT repair in patients with a preoperative SIFK are warranted. Moreover, the current results may enable the determination of cutoff values of all SIFK characteristics, which may provide orthopedic surgeons with guidance during surgical decision-making.

## 5. Limitations

Our study has several limitations. First, this was a retrospective study with a relatively small sample size. Hence, we could not appropriately control for some comorbidities and confounding factors. Second, the included patients’ follow-up duration was short; as such, only the preliminary clinical results were assessed. Finally, the size and length ratios of all SIFK lesions were measured manually; nevertheless, all patient identifiers were completely removed to enable single-blind analysis.

## 6. Conclusions

Preoperative SIFK was noted to be a positive predictor of increased knee OA progression incidence in patients with an MMPRT. Moreover, the lesion size, in terms of the length ratio (in both the sagittal and coronal views on MRI), was positively correlated with knee OA progression severity.

## Figures and Tables

**Figure 1 diagnostics-13-03532-f001:**
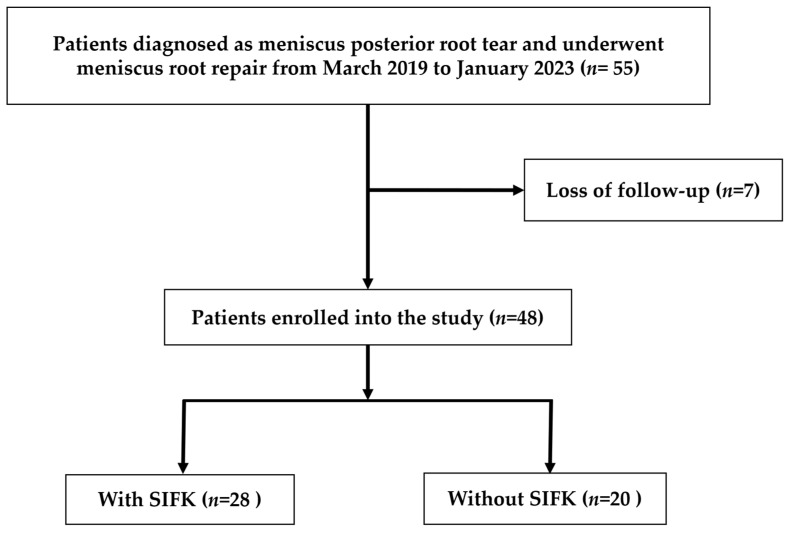
Flowchart of patient inclusion, exclusion, and group stratification.

**Figure 2 diagnostics-13-03532-f002:**
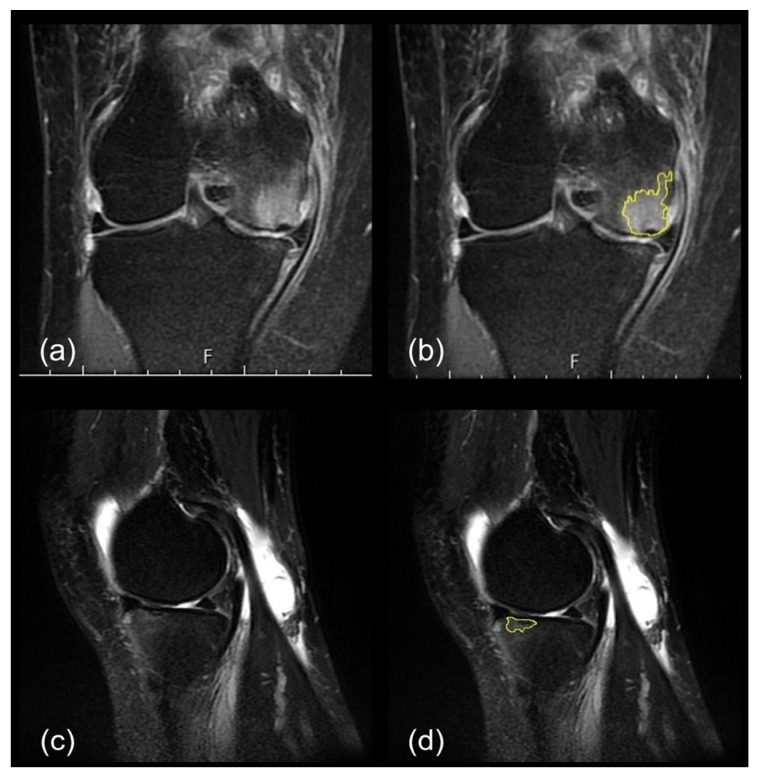
Subchondral insufficiency fracture was identified on medial femoral condyle in the coronal view images (**a**). Depressed part of the subchondral plate and the region exhibiting bone marrow edema were chosen for analysis (**b**). In sagittal view, hyperintense signal was found at anterior portion of tibial plateau (**c**). The thickened subchondral plate with a depressed part in the sagittal view was selected, but the ill-defined area was excluded (**d**).

**Figure 3 diagnostics-13-03532-f003:**
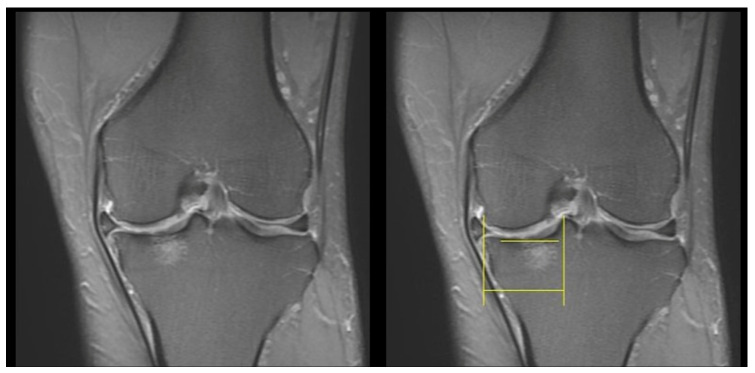
Measurement of the lesion length on coronal view images. Upper horizontal yellow line indicates the horizontal lesion length, and the lower horizontal yellow line indicates the horizontal distance from the medial condyle to the adjacent peak of the tibial spine.

**Figure 4 diagnostics-13-03532-f004:**
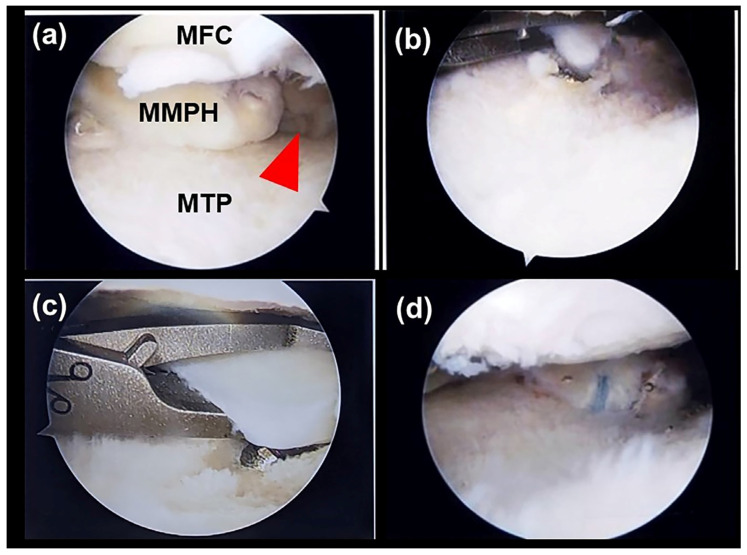
ATPR for an MMPRT in a 48-year-old man. (**a**) Evaluation of the tear pattern and gap of the meniscal root tear (red arrowhead). (**b**) Creation of transtibial tunnel using an aiming drill guide. (**c**) Torn meniscal root sutured using a No. 0 nonabsorbable braided suture with a knee Scorpion needle. (**d**) Reduction of the meniscus to the footprint site by tensioning the free ends of the sutures. MFC, medial femoral condyle; MMPH, medial meniscus posterior horn; MTP, medial tibial plateau.

**Figure 5 diagnostics-13-03532-f005:**
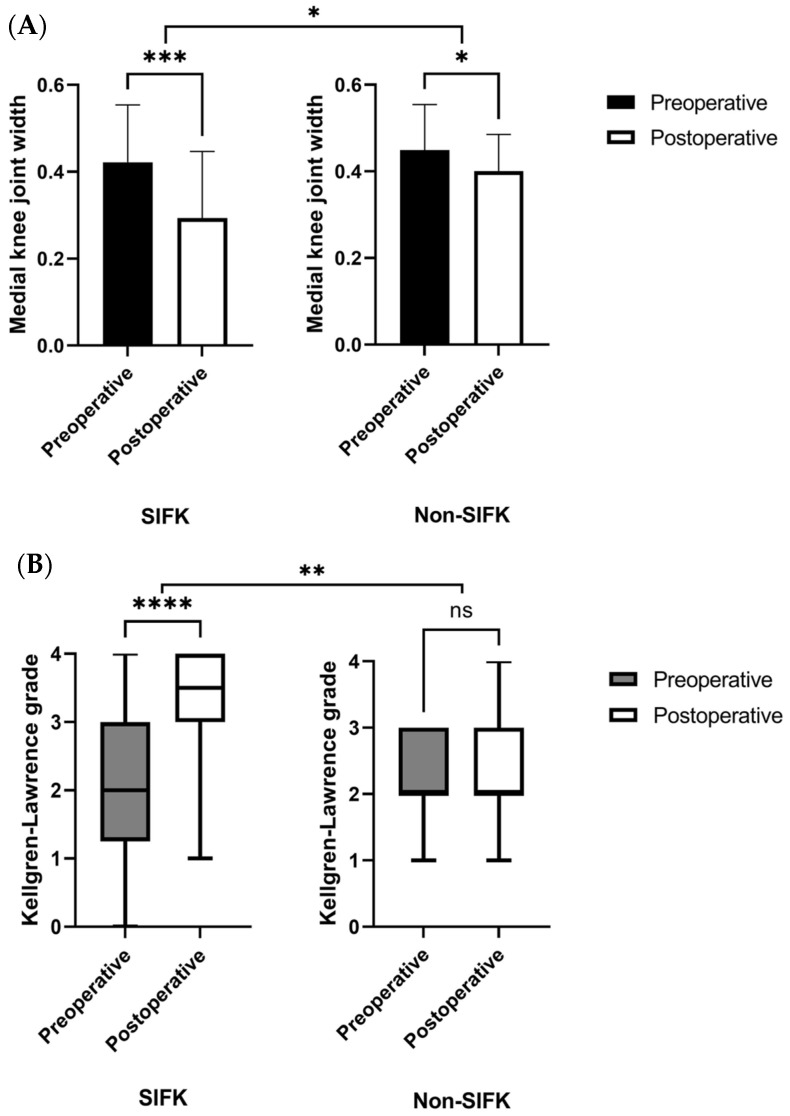
Changes in medial knee joint width (**A**) and alterations in the median KL grades (**B**) postoperatively in each group, assessed using paired *t* and Wilcoxon signed-rank tests, respectively. Medial knee joint width and median KL grade improvements were compared between the two groups using independent-sample *t* and Mann–Whitney *U* tests, respectively. * *p* < 0.05; ** *p* < 0.01; *** *p* < 0.001; **** *p* < 0.0001, ns—not significant.

**Figure 6 diagnostics-13-03532-f006:**
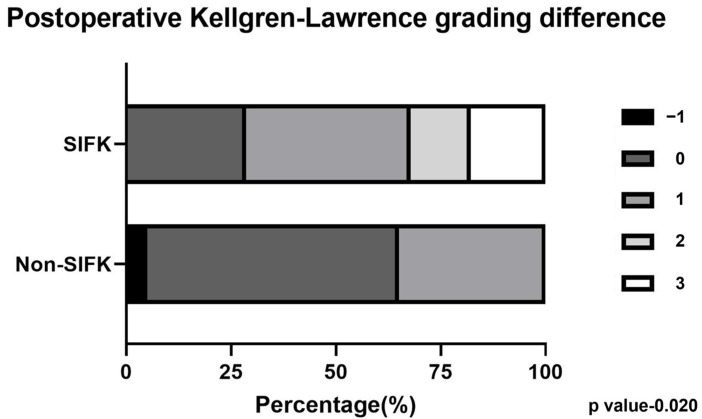
Chi-square test to compare Kellgren–Lawrence grade changes between the SIFK and non-SIFK groups.

**Table 1 diagnostics-13-03532-t001:** Preoperative demographics and clinical characteristics.

Preoperative Characteristics (*n* = 48)	SIFK		*p*	Total
	No (*n* = 20)	Yes (*n* = 28)		
Mean age (year) ^a^	58.34 ± 7.46	58.85 ± 11.52	0.867 ^c^	58.64 ± 9.94
Height (cm) ^a^	159.85 ± 8.60	160.05 ± 9.40	0.939 ^c^	159.97 ± 8.98
Weight (kg) ^a^	72.98 ± 14.68	70.30 ± 14.70	0.536 ^c^	71.41 ± 14.60
Body Mass Index (kg/m^2^) ^a^	28.72 ± 5.58	27.41 ± 5.04	0.398 ^c^	27.95 ± 5.25
Sex (Female/Male) ^b^	14/6	18/10	0.763 ^d^	32/16
^ǂ^ Preoperative medial knee joint width (cm) ^a^	0.45 ± 0.10	0.42 ± 0.13	0.440 ^c^	0.43 ± 0.12
* Preoperative KL grade ^b^	0/4/9/7/0	2/5/13/7/1	0.789 ^d^	2/9/22/14/1
^#^ Follow-up duration (months) ^a^	11.25 ± 9.13	10.74 ± 6.88	0.826 ^c^	10.96 ± 7.80

Note: Two-tailed *p* < 0.05 indicates significance. ^a^ Values are expressed as mean ± standard deviation. ^b^ Values are expressed as grades 0–4. ^c^ Independent-sample *t* test. ^d^ Fisher exact test. ^ǂ^ Preoperative medial knee joint width. * Preoperative KL grade evaluated on the last standing knee AP radiographs preoperatively. ^#^ Follow-up duration: duration between the dates of the latest preoperative and postoperative knee AP standing radiography follow-ups.

**Table 2 diagnostics-13-03532-t002:** Lesion characteristics in different subgroups.

		Difference in Kellgren–Lawrence Grade				*p*
		0 (*n* = 8)	1 (*n* = 11)	2 (*n* = 6)	3 (*n* = 3)	
Coronal view	Femur	1.13 ± 1.88	0.56 ± 0.56	0.43 ± 0.63	0.71± 0.94	0.196 ^a^
	Tibia	0.79 ± 1.12	1.42 ± 1.17	0.97 ± 0.88	1.55± 0.89	0.008 ^a^
Sagittal view	Femur	1.26 ± 1.65	0.86 ± 1.02	1.14 ± 1.52	0.62 ± 0.87	0.063 ^a^
	Tibia	1.10 ± 1.56	1.47 ± 1.34	1.17 ± 0.91	1.77 ± 1.70	0.031 ^a^
Lesion-to-tibia length ratio		0.30 ± 0.30	0.65 ± 0.37	0.59 ± 0.35	0.83 ± 0.34	0.002 ^a^
Site	Medial femur	6	7	3	3	0.629 ^b^
	Lateral femur	0	2	1	0	0.705 ^b^
	Medial tibia	6	11	6	3	0.233 ^b^
	Lateral tibia	0	0	0	0	- ^c^

Note: Values are expressed as means ± SDs or numbers. In subgroup 2, values are expressed as the absolute measured numbers because only one participant was enrolled. ^a^
*p* < 0.05, Kruskal–Wallis test. ^b^
*p* < 0.05, Fisher exact test. ^c^ Could not be computed for being constant.

## Data Availability

The data presented in this study are available on request from the corresponding author. The data are not publicly available due to privacy.

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
