# Peer review of "Correlation between Subchondral Insufficiency Fracture of the Knee and Osteoarthritis Progression in Patients with Medial Meniscus Posterior Root Tear"

_diagnostics, 2023, doi:10.3390/diagnostics13233532_

Round 1
Reviewer 1 Report
Comments and Suggestions for Authors
Title: report the type of study
Abstract:
purpose is too long. only one sentence please
methods: not clear. report clear methods of the study
results: report precise p-value and results. these are not results
conclusions: can be reported without clear results
Introduction: little bit too long. focus on the aim of your paper. report what is known and what is unknown highlighting the controversies. explain the ratio for your study. finish with aim and hypothesis
methods: strobe checklist are missing
you report only exclusion criteria. add also inclusion criteria
who performed surgery?
results: well reported
discussion
start with main findings of your paper. report what is new and what add to current literature
analyze the clinical impact of your study, how can improve clinical daily practice?
limitations: much more than those reported
conclusions
coherent
references: add following
Serrano DV, Saseendar S, Shanmugasundaram S, Bidwai R, Gómez D, D'Ambrosi R. Spontaneous Osteonecrosis of the Knee: State of the Art. J Clin Med. 2022 Nov 25;11(23):6943. doi: 10.3390/jcm11236943. PMID: 36498517; PMCID: PMC9737125.
D'Ambrosi R, Meena A, Raj A, Ursino N, Mangiavini L, Herbort M, Fink C. In elite athletes with meniscal injuries, always repair the lateral, think about the medial! A systematic review. Knee Surg Sports Traumatol Arthrosc. 2023 Jun;31(6):2500-2510. doi: 10.1007/s00167-022-07208-8. Epub 2022 Nov 2. PMID: 36319751; PMCID: PMC10183423.
Comments on the Quality of English Languagecan be improved
Author Response
Dear respected reviewer:
We sincerely appreciate the opportunity to revise our manuscript based on the valuable feedback provided by the reviewers and editors. We have taken careful consideration of the comments and suggestions, and as a result, we have made substantial improvements to the manuscript to ensure its alignment with the journal's scope and standards.
1. Author responses: We have made changes according to your suggestions.
- The purpose of this study has been rewritten. (line 28-29)
- The method section has been revised. (line 30-34)
- Precise p-values were added. (line 34-40)
2. Introduction: little bit too long. focus on the aim of your paper. report what is known and what is unknown highlighting the controversies. explain the ratio for your study. finish with aim and hypothesis
Author responses: The introduction part has been summarized to be more focus on issues relevant to current study. In the final section, the hypothesis of this study was emphasized. (line 90-93)
3. methods: strobe checklist are missing
Author responses: Strobe checklist has been attached as supplementary file.
4. you report only exclusion criteria. add also inclusion criteria
Author responses: The inclusion criteria of current study were added. (line 101-105)
5. who performed surgery?
Author responses: All surgeries were performed by a single orthopedic surgeon (C-C.W.) who is experienced in arthroscopic surgery and sports medicine followed standard operative procedures. (line 177-179)
6. results: well reported
7. discussion
start with main findings of your paper. report what is new and what add to current literature. analyze the clinical impact of your study, how can improve clinical daily practice?
Author responses: the discussion part hasd been rewritten. The main findings were stated in the very first beginning of dicussion. (line 289-293). Moreover, the clinical signficance of this study was emphasized, please kindly refer to line 314-318; line 345-349; line 371-375.
limitations: much more than those reported
Author responses: The discussion part has been rewritten, and be more focusing on reviewing the similarities and difference between literature and current study. The impact and novelty of main finding of this study was emphasized. (line 303-309). The limitations of this study were rewritten to be more comprehensive and precise. (line 350-362)
conclusions
coherent
references: add following
Serrano DV, Saseendar S, Shanmugasundaram S, Bidwai R, Gómez D, D'Ambrosi R. Spontaneous Osteonecrosis of the Knee: State of the Art. J Clin Med. 2022 Nov 25;11(23):6943. doi: 10.3390/jcm11236943. PMID: 36498517; PMCID: PMC9737125.
D'Ambrosi R, Meena A, Raj A, Ursino N, Mangiavini L, Herbort M, Fink C. In elite athletes with meniscal injuries, always repair the lateral, think about the medial! A systematic review. Knee Surg Sports Traumatol Arthrosc. 2023 Jun;31(6):2500-2510. doi: 10.1007/s00167-022-07208-8. Epub 2022 Nov 2. PMID: 36319751; PMCID: PMC10183423.
Author responses: those references have been added.
Reviewer 2 Report
Comments and Suggestions for Authors
Dear Author's
Thank you for allowing me to read the results of the submitted article. I think that the article is interesting. In my opinion, it requires a few minor additions, namely: 1. please specify the conclusion in summary, at this stage it sounds like a hypothesis, this should be a clear conclusion 2. I suggest reviewed article by native speaker 3. I don't see the approval of the bioethics committee, do the authors have one?best regards for all Author's
Comments on the Quality of English Language I suggest reviewed article by native speaker.
Author Response
Dear respected reviewer:
We sincerely appreciate the opportunity to revise our manuscript based on the valuable feedback provided by the reviewers and editors. We have taken careful consideration of the comments and suggestions, and as a result, we have made substantial improvements to the manuscript to ensure its alignment with the journal's scope and standards.
Dear Author's
Thank you for allowing me to read the results of the submitted article. I think that the article is interesting. In my opinion, it requires a few minor additions, namely: 1. please specify the conclusion in summary, at this stage it sounds like a hypothesis, this should be a clear conclusion 2. I suggest reviewed article by native speaker 3. I don't see the approval of the bioethics committee, do the authors have one?
best regards for all Author's
- please specify the conclusion in summary, at this stage it sounds like a hypothesis, this should be a clear conclusion 2. I suggest reviewed article by native speaker 3. I don't see the approval of the bioethics committee, do the authors have one?
Dear respected reviewer:
Author responses:
- The conclusion part has been rewritten to be more precise and comprehensive. (line 363-366)
- The manuscript has been proofread by academic English editing services (Wallace academic).
- The IRB approval letter was attached as supplementary file.
Round 2
Reviewer 1 Report
Comments and Suggestions for Authors
accept